# Defining a New NLP Playground

**Sha Li[1], Chi Han[1], Pengfei Yu[1], Carl Edwards[1], Manling Li[2], Xingyao Wang[1],**
**Yi R. Fung[1], Charles Yu[1], Joel R. Tetreault[3], Eduard H. Hovy[4], Heng Ji[1]**

[1] University of Illinois Urbana-Champaign  [2] Northwestern University
[3] Dataminr, Inc.  [4] University of Melbourne

{shal2, chihan3, pengfei4, cne2, xingyao6, yifung2, ctyu2, hengji}@illinois.edu

manling.li@northwestern.edu, jtetreault@dataminr.com, eduard.hovy@unimelb.edu.au

## Abstract

The recent explosion of performance of large language models (LLMs) has changed the field of Natural Language Processing (NLP) more abruptly and seismically than any other shift in the field's 80-year history. This has resulted in concerns that the field will become homogenized and resource-intensive. The new status quo has put many academic researchers, especially PhD students, at a disadvantage. This paper aims to define a new NLP playground by proposing 20+ PhD-dissertation-worthy research directions, covering theoretical analysis, new and challenging problems, learning paradigms, and interdisciplinary applications.

## 1 Introduction

It is the best of times. It is the worst of times. We are living in an incredibly exciting yet strange era of Natural Language Processing (NLP) research due to the recent advancements of large language models (LLMs) on various data modalities, from natural language (Brown et al., 2020) and programming language (Chen et al., 2021; Wang et al., 2023a) to vision (Radford et al., 2021; Li et al., 2022a; Wang et al., 2022b) and molecules (Edwards et al., 2022; Zeng et al., 2022; Su et al., 2022).

At the core, LLMs produce text sequences word-by-word by computing conditional probability based on context. At a sufficiently large scale, they can answer questions, generate arguments, write poetry, impersonate characters, negotiate contracts and achieve competitive results across a wide variety of standard NLP tasks including entity typing, sentiment analysis, and textual entailment, showcasing "emergent behavior" such as in-context learning (Wei et al., 2022).

However, this "moment of breakthrough" received a polarized response in the NLP research community: while some welcomed the progress, others felt lost. Why is NLP so vulnerable to a single advancement?

In retrospect, when NLP adopted the machine learning paradigm in the early 1990s it started along a journey that led to increased homogeneity. The dominant methodology became: (1) Identify a challenge problem or task; (2) Create a dataset of desired input-output instances; (3) Select or define one or more evaluation metrics; and (4) Develop, apply, and refine machine learning models and algorithms to improve performance.

If a challenge did not support the creation of a dataset (e.g., text styles of people in different professions) or metric (e.g., summaries of novels or movies), or worse yet if it was not amenable to a machine learning solution, then mainstream NLP simply did not address it. For a long time, NLG was in this position because its starting point —semantic representations— were neither standardized, nor easy to produce at scale, nor amenable to direct evaluation. No dataset, no metric — little attention. Yet multi-sentence NLG starting with deep semantic input, and with output tailored to different audiences, is arguably the most complex task in NLP, since it involves so many aspects of linguistic communication together. As such, it surely deserved the concentrated effort that NLP has bestowed on MT, Speech Recognition, QA, and other major challenges in the past.

Suddenly, within the space of a few months, the landscape changed. NLP encountered an engine that seemingly could do everything the field had worked on for decades. Many subtasks in NLP seemed to become irrelevant overnight: Which grammar formalism to parse into? Which rhetorical structure and focus control model for multi-sentence coherence? Which neural architecture is optimal for information extraction or summarization? None of that matters if the magical engine can do the entire end-to-end language-to-language task seamlessly (Sanh et al., 2022; OpenAI, 2023).

Dozens of Ph.D. theses lost their point, because their point was a small step in the process that no longer seemed needed. The dominant paradigm is also challenged: instead of setting up benchmarks and then developing models accordingly, people started discovering new abilities of such models (Bubeck et al., 2023) (who knew that LLMs could draw unicorns using TikZ?).

An important constraint is the practicality of the goal. This newer generation of LLMs is beyond the practical reach of all but a small number of NLP researchers. Unless one of the organizations building LLMs provides free access for research —an unlikely occurrence given the estimated six-figure monthly expense to run one— or a procedure is developed to construct university-sized ones cheaply, the academic NLP community will have to be quite creative in identifying things that either generative LLMs cannot do *in principle* or applications that can be built without re-training them and at the same time are important and doable *in practice*.

Inspired by the efforts of a group of PhD students (Ignat et al., 2023), we believe it would be a valuable exercise to define new research roadmaps. We believe that while LLMs seemingly close research avenues, they also open up new ones. Current LLMs remain somewhat monolithic, expensive, amnesic, delusional, uncreative, static, assertive, stubborn, and biased black boxes. They still have a surprising deficiency (near-random performance) in acquiring certain types of knowledge (Wang et al., 2023f), knowledge reasoning and prediction. In this paper, we aim to define a new NLP playground by proposing a wide range of PhD-dissertation-worthy research directions to democratize NLP research again. In particular, we cover observations and suggestions along the perspectives of LLM theory (Section 2), challenging new tasks (Section 3), important but understudied learning paradigms (Section 4), proper evaluation (Section 5), and interdisciplinary applications (Section 6).

## 2  Theoretical Analysis of LLMs

There is a growing necessity to open the black box of machine learning models through theoretical analysis. In this section, we advocate for both **mathematical** (by mathematical analysis) and **experimental** (inducing rules and laws such as Ghorbani et al. (2021); Hoffmann et al. (2022) from extensive experimental observations) theories of LLMs.

### 2.1  Mechanism Behind Emergent Abilities

LLMs have displayed impressive emergent capabilities such as instruction following, chain-of-thought reasoning, and in-context learning (Brown et al., 2020; Wei et al., 2022; Min et al., 2022; Wei et al.; Logan IV et al., 2022; Wei et al., 2021). For example, the ability of **instruction following** enables models to follow novel instructions. For guidance on prompting beyond heuristics, we need a comprehensive understanding of how instructions work. Some initial theories suggest an explanation through Bayesian inference (Jiang, 2023), which relies on strong assumptions without practical insights. Here we advocate for theories on the feasibility of constraining or measuring models' deviation from instructions. A multi-player setting is also important, where one user's prompt is composed with another player's prompt (such as OpenAI's hidden meta instruction) before being fed into LLMs, where additional security issues might arise for the first user.

**Chain-of-thought (CoT)** reasoning is where LLMs tackle complex tasks by generating solutions in a sequential, step-by-step manner. CoT theoretically enhances the computational capacity of Transformer-based models to solve problems exceeding $\mathcal{O}(n^2)$ complexity. While some constructive explanations have been suggested (Feng et al., 2023a), they are not fully validated as the underlying mechanism. Importantly, it is worth investigating the verifiability problem of the reasoning chain (whether CoT can be trusted as a valid logic chain) and its calibration (whether LLMs formulate ad-hoc CoTs for arbitrary conclusions).

**In-context learning (ICL)**, where LLMs learn from demonstration examples in-context without parameter updates, has seen explanations based on gradient-descent (Akyürek et al., 2022; von Oswald et al., 2022), kernel regression (Han et al., 2023a) or Bayesian inference (Xie et al.; Jiang, 2023; Wang et al., 2023d). Important challenges remain and necessitate more comprehensive explanations, such as sensitivity to example order and robustness to perturbed input-output mapping. We hypothesize that a deeper understanding of how LLMs balance algorithmic solutions with implicit language inference can help clarify these questions, which might be approachable by exploring how LLMs disentangle semantic and functional information.

**Model-specific vs. Model-agnostic** is a persistent gap among explanations, raising the question of whether the emergent abilities depend on the Transformer architecture or simply fitting the pre-training data. With some recent work suggesting that other architectures achieve comparable performance in some domains (Peng et al., 2023; Zhai et al., 2021), this open question is important for prioritizing among model design (including other architectures), prompting engineering, and simply carefully collecting larger datasets. To bridge this gap, we also advocate for theoretical frameworks beyond (mixture) of HMMs to better model language data properties.

## 2.2 Theoretical Robustness and Transparency

**Robustness** is to ensure that no backdoor designs or adversarial usages can be easily implemented in the model. Although not a novel problem by definition, this issue has new implications and formulations in the LLM era. In a situation where most users do not have access to the pre-training and model-editing details, we call for research into robustness diagnosis *for arbitrary given LLM*. Despite negative evidence suggesting it may be nearly impossible to prevent adversarial prompting under certain conditions (Wolf et al., 2023), we maintain a positive outlook and hope that it can be potentially overturned under more realistic conditions, such as high computational complexity in searching for adversarial prompts.

**Transparency** in LLMs is concerned with alignment between the model's self-explanations and its internal computational rationale. With empirical studies suggesting that LLMs may not always accurately express their "thoughts" (Turpin et al., 2023), computational modeling of LLM intentions becomes essential. The quest for transparency is important for preventing LLMs from generating misleading rationales to humans. We advocate for establishing both positive and negative theorems on counteracting false rationales under different conditions, along with examining associations between "faithfulness" modes and neuron activities in specific architectures.

## 3 New and Challenging Tasks

### 3.1 Knowledge Acquisition and Reasoning

**Knowledge inside LLMs** The black box property of LLMs poses a significant challenge when it comes to evaluating implicit knowledge within the model. Initial studies have been conducted to elicit/identify (Cohen et al., 2023; Shin et al., 2020; Petroni et al., 2019, 2020; Fung et al., 2023; Gudibande et al., 2023; Li et al., 2023c) and localize/edit knowledge (Dai et al., 2021; Meng et al., 2022a,b; Zhu et al., 2020; Mitchell et al., 2022a; De Cao et al., 2021; Hase et al., 2023; Meng et al., 2022a; Mitchell et al., 2022b). However, our understanding of the knowledge organization within language models (*where* and *how* knowledge is stored) is still limited, and it remains uncertain whether full comprehension is achievable. Moreover, existing studies primarily focus on factual or commonsense knowledge, overlooking more complex knowledge such as rules of inference (Boolos et al., 2002).

**Large-Scale Knowledge Reasoning** LLMs have demonstrated promising performance across various reasoning tasks (Dua et al., 2019; Miao et al., 2020; Cobbe et al., 2021; Yu et al., 2020; Bhagavatula et al., 2020; Talmor et al., 2019) when appropriately prompted, such as through the use of Chain-of-Thought and improved Chain-of-Thought (Wei et al.; Chowdhery et al., 2022; Xue et al., 2023; Diao et al., 2023; Wang et al., 2023e; Paul et al., 2023) or Program-of-Thought (Chen et al., 2022). However, current reasoning benchmarks (Cobbe et al., 2021; Ling et al., 2017; Patel et al., 2021; Hosseini et al., 2014; Miao et al., 2020; Koncel-Kedziorski et al., 2016; Talmor et al., 2019; Geva et al., 2021) focus on reasoning with small-scale context, typically consisting of hundreds of words. This level of reasoning falls short when tackling complex tasks, such as scientific research, which demands knowledge from extensive volumes of related literature and domain-specific knowledge bases. Retrieval-augmentation (Guu et al., 2020; Khandelwal et al., 2020; Borgeaud et al., 2022; Izacard et al., 2022; Lai et al., 2023b) serves as a powerful tool for integrating large-scale contextual knowledge into language models. However, current retrieval methods predominantly rely on semantic similarities, while humans possess the *accommodative* learning (Illeris, 2018) ability to draw inspirations from semantically dissimilar knowledge and transfer it to the target task. To achieve this, we not only need to extend the input context length, but also understand how models organize knowledge and develop more effective knowledge representations and evaluation metrics (Section 5).

**Faithfulness and Factuality**   Ensuring the truthfulness of generation output requires optimal utilization of internal knowledge within the model and external knowledge, which includes the input context, knowledge bases, and open web resources. Access to external knowledge typically relies on the success of information retrieval (Lewis et al., 2020; He et al., 2023; Yu et al., 2023c,b), information extraction (Wen et al., 2021; Huang et al., 2023), grounded generation (Li et al., 2021, 2022b; Gao et al., 2023a; Weller et al., 2023; Lai et al., 2023a) and knowledge-augmented generation (Petroni et al., 2020; Geva et al., 2023). Internal knowledge involves the implicit parametric knowledge stored within the model, the correction and refinement of which is limited to the inference stage (Lee et al., 2022; Meng et al., 2022a,b; Chen et al., 2023a). To effectively minimize hallucination and correct factual errors, it is crucial to not only decipher how knowledge is interpreted through model parameter patterns, but to understand how the model pieces knowledge together and governs the underlying logic during generation. A significant challenge in knowledge-guided generation is defining an appropriate knowledge representation that supports both complex structures and distributed representations. We believe this representation should combine the strength of symbolic-based reasoning to minimize unwarranted inferences, and the flexibility of distributed representations to encode any semantic granularity. Drawing insights from misinformation detection and knowledge comparative reasoning systems could also be one useful dimension of signals for improving faithfulness and factuality (Liu et al., 2021a; Fung et al., 2021; Wu et al., 2022, 2023).

## 3.2   Creative Generation

Although people have long envisioned using models for creative writing, this has only become a reality recently, when language generation models could reliably produce fluent text. Compared to previous sections where generated text is a vehicle for knowledge, creative use cases focus more on the style or form of language and encourage open-ended output. [1]

**Creative Writing Assistance**   Since language models offer conditional generation ability out-of-

the-box, they have been adopted by many people in the creative industry for brainstorming or research tools (Kato and Goto, 2023; Gero et al., 2023; Halperin and Lukin, 2023). One key challenge for such tools is promoting creative generation, instead of generating the most probable continuation, which was what language models were trained for. Current LMs have been observed by writers to over-rely on cliques or tropes and produce overly moralistic and predictable endings (Chakrabarty et al., 2024). While the plot should be unexpected, details in the story should not go against commonsense (unless it is part of the setting), and maintain consistency within the story. This requires a model that enables controllability over the level of creativity in its output. Do we need to train a more creative model, or can we fix the problem at the inference stage? On the other hand, the focus on detoxification of LMs through RLHF (reinforcement learning with human feedback) might have led to the incompetency of the model in navigating deeper and morally challenging themes.

Another direction for exploration is how to build better writing tools that work together with humans. Some attempts have been made to allow users to interact through instructions (Chakrabarty et al., 2022) or use editing sequences to improve writing quality (Schick et al., 2022). These could serve as critical building blocks toward the goal of developing a model that supports different types of input and can improve itself and personalize through interaction. In addition, models can also assist in different stages of writing, such as world-building and reviewing drafts. It remains to be explored where the model is most effective and where human writers should step in and make decisions.

**Interactive Experiences**   Text generation models can not only be assistants for writing static scripts but also open up an opportunity to create dynamic and personalized experiences for the user by conditioning on their input. These interactive experiences can be used for education, therapy, game design, or filmmaking. More recently, there have been attempts to connect conversational models with other components such as speech recognition, text-to-speech, and audio-to-face rendering to create an end-to-end immersive experience of interacting with non-playable characters[23]. Another related open area for exploration is to create

---

[1]In this section we limit our scope to applications of text generation, however, we fully acknowledge the potential of multi-modal creative generation, such as generating personal avatars, movie clips, and 3D scenes.

[2]NVIDIA blog
[3]https://charisma.ai/

emotion-oriented experiences, which is one of the key goals of storytelling (Lugmayr et al., 2017). We should consider creating narratives based on the desired emotional response and the reader's feedback (Brahman and Chaturvedi, 2020; Ziems et al., 2022; Mori et al., 2022).

## 4   New and Challenging Learning Paradigms

### 4.1   Multimodal Learning

In light of the remarkable progress of the language world, we are now poised to venture into a multitude of modalities that were previously beyond consideration. Some learning signals stem from reading static data, such as images, videos, speech, and more, which will be discussed in this section; while other signals require interacting with the physical world, which will be detailed in Section 4.2.2.

Multimodal encoding, at its core, involves learning the "correspondence" or "alignment" among various modalities, which always facing the challenges of **Granularity Difference** across modalities. This is a new and growing area with several solutions proposed to align across modalities: (1) a hard alignment that enables granularity-aware fusion (Tan and Bansal, 2020; Li et al., 2022a; Momeni et al., 2023; Wang et al., 2022c, 2023f); (2) a soft alignment to project the text space with the vision space (Zhou et al., 2023; Li et al., 2023b; Zhu et al., 2023; Lin et al., 2023). Beyond these semantic alignment challenges, there are further difficulties when it comes to non-semantic abstractions:

**Geometric Reasoning:** Recognizing spatial relationships, such as "*left*", "*right*", "*beside*", "*above*", or "*behind*", requires comprehensive geometric mental simulation, which existing models consistently making errors (Kamath et al., 2023). Maintaining transformation invariance, regardless of position, rotation, or scale, remains a core challenge. Besides, current models, predominantly trained on 2D images, inherently miss out on the intricacies of 3D spatial configurations, inhibiting understanding of depth and relative object sizes based on distance. To address these challenges, existing efforts augment existing large models with an agent view to infer spatial layouts, predicting possible navigations from visual and textual cues (Liu et al., 2022; Berrios et al., 2023; Feng et al., 2023b). However, we believe the underlying challenge lies in the missing objective of geometric reasoning. Existing

pretraining paradigms predominantly focus on semantic alignment between image/video-language pairs, while features (e.g., low-level edges, lines) are largely omitted in the encoded image representation.

**Context Ambiguity:** Accurate understanding should factor in the wide context of temporal dynamics, social dynamics, emotional dynamics, and more. The temporal dimension presents a unique challenge in understanding vision and speech. Existing methods only focus on temporal ordering (Zellers et al., 2021, 2022) and forward/backward generation (Seo et al., 2022; Yang et al., 2023a; Cheng et al., 2023). However, temporal dynamics is much more complicated. For instance, a video gesture (like a nod) may correspond to a later affirmation in the speech (Li et al., 2019). Such ambiguity requires reasoning over a wider context with various constraints. Emotion, another yet-underexplored abstract dimension, is conveyed through tone, pitch, speed in speech, and through expressions or body language in vision. Besides, social norm understanding is challenging as the same word or facial expression can convey different emotions depending on the context. Thus, potential solutions require to take into account various contexts, including preceding conversations or events, along with casual reasoning.

**Hierarchical Perception:** Human cognition is inherently hierarchical. When processing visual signals, our attention is not uniformly distributed across every pixel but focus on salient regions that carry the most information, allowing us to quickly identify key features and make sense of our surroundings (Hochstein and Ahissar, 2002; Eickenberg et al., 2017). However, existing models overlook such attention hierarchy and tend to lose focus when asking about visual details (Gao et al., 2023b). To address this challenge, interpreting natural scenes requires hierarchical recognition, from broader contexts down to detailed attribute abstraction. Besides, aligning visual hierarchies with linguistic structures is important. Further, it requires the ability to perform abstraction over details, balancing between an abstracted scene understanding and intricate recognition is an ongoing challenge.

### 4.2   Online Learning

Trained on static corpora, existing models are incapable of keeping themselves updated on new information or learning from interaction history for

self-improvement. To alleviate these issues, this section discusses the need for next-generation models to learn in an *online* setting.

### 4.2.1 Updating Information Within Models

A straightforward approach to updating models is to continue training on new data. This is however not efficient, since we only care about new information which accounts for a small fraction of the data, nor effective, as fine-tuning on new data might interfere with learned information in models. To achieve efficient updates, we would like the model to automatically identify notable information in new data (Yu and Ji, 2023) instead of relying on heavy human selection or preprocessing as in knowledge editing tasks (Dai et al., 2021; Meng et al., 2022a,b; Zhu et al., 2020; De Cao et al., 2021; Hase et al., 2023; Mitchell et al., 2022b). Effectively updating the model requires overcoming the bias toward (Yu and Ji, 2023; Wei et al., 2023) as well as avoiding catastrophic forgetting (McCloskey and Cohen, 1989; Ratcliff, 1990) of learned prior information. This might be achieved by changing the training paradigm to increase model capacity over time (e.g. progressive training (Gong et al., 2019), MoE (Shen et al., 2023)) or better understanding of knowledge organization within models (as detailed in Section 3.1) so that edits can be performed with minimal interference.

### 4.2.2 Learning from Continuous Interactions

Interaction is essential in human learning (Jarvis, 2006). Humans learn how to best tackle different tasks by interacting with the **environment**, and they learn social norms from their interactions with other **humans**. Moreover, such interactions are **multi-turn** in nature, allowing humans to iteratively refine their actions for the task at hand *and* continuously improve their mental model's capability of performing similar tasks in the future.

**Interaction with Environments**  We consider environments a broad category of systems that provide feedback upon actions. The world we live in can be regarded as a typical environment: the law of physics would decide the world state change and provide sensor stimuli to the actor (e.g., Ahn et al. (2022)). Training a model (i.e., Embodied AI) that can interact with the physical world through multi-modal input (Driess et al., 2023; Jiang et al., 2023) poses challenges related to multi-modal learning (Section 4.1) as well as unique challenges due to long-horizon planning requirements and dynamic environments. The concept of environments also extends to human-crafted environments (e.g., programming language interpreters (Wang et al., 2023b), embodied simulators (Shridhar et al., 2020)) that provide automated feedback for any input by rules. Such artificial environments allow easy collection of automatic feedback which could prepare models for deployment in the physical world.

**Interaction with Humans**  Beyond learning from generic human preference towards building generalist agents (Ouyang et al., 2022), real-world applications typically require customizable solutions (e.g., personalized agents) to be created efficiently. We advocate for a new learning paradigm where models can be taught through (multi-modal) interactions with humans, including natural language feedback (Padmakumar et al., 2022; Wang et al., 2023c) and physical demonstration (Lee, 2017). Such complex problem nature may also involve customized retrieval from a large toolset of specialized models and effective action planning (Qin et al., 2023; Yuan et al., 2023).

## 5 Evaluation

As models become increasingly powerful and multi-purpose, their evaluation has become a growing bottleneck for advancing NLP. We first discuss the question of "what should be evaluated" followed by "how should we measure performance."

### 5.1 Benchmarks

Language models are known to be multi-task learners, and the new generation of LLMs can achieve impressive performance under few-shot or even zero-shot conditions. This has led to the creation of many general benchmarks such as GLUE (Wang et al., 2018), SuperGLUE (Wang et al., 2019), MMLU (Hendrycks et al., 2021), Super-NaturalInstructions (Wang et al., 2022a), HELM (Liang et al., 2022), and AGIEval (Zhong et al., 2023). While setting up comprehensive benchmarks is useful, current benchmarks still have the following limitations: (1) lack diverse and difficult tasks that are important for real-world applications; (2) only contain static data sets that are not sufficient for applications that require multi-turn context-dependent input such as situation-grounded dialog; (3) robustness deficiencies, and (4) lack of support for performance analysis.

Although some benchmarks extend to thousands of NLP tasks, most of them are variants of sentence-level tasks, while ignoring more challenging tasks such as structured prediction and cross-document reasoning. For example, Li et al. (2023a) reported that LLMs methods obtained 25.2%-68.5% lower performance than state-of-the-art methods based on much smaller models for nearly all of the Information Extraction tasks. Task design should also aim to assist with human users' daily tasks, as exemplified by the most popular tasks being related to planning and seeking advice by the ChatGPT users at ShareGPT [4]. Another issue is that benchmarks quickly saturate due to the development of newer models, and thus "live" benchmarks that can be updated over time (Kiela et al., 2021) might be worth pursuing.

To move beyond static data, we believe that simulated environments such as large-scale multi-player game environments can serve as an efficient solution. Games have been used as a way of benchmarking progress of reinforcement learning algorithms (Silver et al., 2018; Guss et al., 2021) and also used to collect static datasets in NLP (Urbanek et al., 2019; Bara et al., 2021; Lai et al., 2022). Game worlds provide a cheap way to explore different environments and situations, which is necessary for grounded language learning and learning through interaction. Humans can interact with models playing as characters in the game to evaluate their performance, or we can let models interact with each other (Park et al., 2023) and evaluate their interaction behavior as a whole.

Finally, we advocate for work on model diagnosis beyond the current brittle paradigm of case study through manual inspection: methods that help identify which parts of the input the model underperform on (Liu et al., 2021b), what are the model's behavior patterns and what data this performance could be attributed to (Ilyas et al., 2022).

## 5.2 Metrics

Automatic evaluation metrics have been an accelerant for NLP progress in the last 20 years. Heuristic-based metrics (Papineni et al., 2002; Lin, 2004; Lavie and Agarwal, 2007) have been found to correlate weakly with human preferences (Liu et al., 2016). As a result, the field has pivoted to model-based metrics which have shown better alignment with human judgment (Lowe et al., 2017; Zhang

et al., 2020; Sellam et al., 2020; Yuan et al., 2021; Zhong et al., 2022). However such metrics might allow for shortcut approaches or come with biases embedded in the scoring model (Sun et al., 2022).

Automatic metrics struggle with open-ended natural language generation problems such as conversation and creative writing tasks due to the absence of ground truth. LLMs present an opportunity to tackle this problem (Zheng et al., 2023; Fu et al., 2023; Liu et al., 2023b), but they also suffer from certain biases including position, verbosity, and self-enhancement biases (models prefer themselves) that users should be cautious about. We need to develop metrics beyond accuracy and evaluate aspects such as robustness (Chen et al., 2023b), bias, consistency (Chan et al., 2023), informativeness, truthfulness, and efficiency.

On the other hand, human evaluation has traditionally been perceived as the more trustworthy evaluation method and a better indicator of the model utility. However, as models improve, it is questionable whether crowdworkers are adequate to serve as assessors (or annotators), particularly in fields such as science, healthcare, or law. Annotator bias (Geva et al., 2019; Sap et al., 2022) and disagreement (Fornaciari et al., 2021) should also be taken into consideration. If we design our models to be "assistants", a more useful human evaluation might not be to identify which output is more correct, but which output can help the human complete the task more efficiently.

## 6 NLP+X Interdisciplinary Applications

### 6.1 Human-Centered NLP

As LLMs become ubiquitous in both the research and public spheres, mitigating potential harms, both allocation and representation (Blodgett et al., 2020), to social groups using these models must be a core consideration. Social bias and stereotypes are a common way for LLMs to materialize these internal defects, so debiasing these models is important for fairness and robustness. Furthermore, LLMs must be aware of the extra-contextual requirement of abiding by the sociocultural norms expected by the user (Fung et al., 2023), especially when used as chatbots directly interacting with humans.

Post-hoc debiasing and improving the social awareness of pretrained LLMs are important to this end. Though modern approaches have made great advances in democratizing LLM training, most

---

[4]https://sharegpt.com/

builders don't have a need to pretrain their own LLMs, opting to, at most, fine-tune them. Rather than hope that an LLM is unbiased after pretraining, many researchers have discussed the utility in having a separate general debiasing step to account for any unintended associations stemming from pretraining (Yu et al., 2023a; Omrani et al., 2023; Yang et al., 2023b). Relatively less explored is the complementary requirement of augmenting LLMs with the awareness and ability to abide by sociocultural norms. The crux of the problem is training the model to recognize *what* behaviors in its training data are the results of sociocultural norms, discover *why* and *when* those norms should be followed, and *how* those norms can be followed (i.e., is it only in a specific way or is this a behavior that can be generalized across situations?).

Another important direction is personalization based on the user, particularly for chatbots. LLMs have an amazing ability to multiplex behavior based on the language context provided in the prompt (Section 2.1), but they do not have the ability to account for the audience apart from what's inferred from text. This poses a problem for personalization because the same context or conversation can have differing levels of appropriateness depending on the audience (e.g., something that one finds relatively harmless may be incredibly offensive to someone else). Thus, we must improve LLMs' ability to infer the personal norms and appropriate behaviors in each individual context independently and act accordingly. This may, in part, involve bridging the gap between distant users who share similar beliefs to decode latent representations (Sun et al., 2023). In parallel, we can also provide users with multi-dimensional controls for generation (Han et al., 2023b), including their sentiment, political stance, and moral values, so that they can directly influence the model's language usage.

## 6.2 NLP for Science

One area with the most potential impact from NLP is science (Hope et al., 2022; Zhang et al., 2023). Although researchers have long been interested in extracting actionable information from the literature (Hersh and Bhupatiraju, 2003; Griffiths and Steyvers, 2004; Li et al., 2016; Wang et al., 2021), this has been challenging due to the variety and complexity of scientific language. With the growing capabilities of NLP techniques, intensified focus is now deserved because of both the potential impacts and the challenges that will need to be overcome.

One exciting emerging area is jointly learning natural language and other data modalities in the scientific domain (Edwards et al., 2021; Zeng et al., 2022; Edwards et al., 2022; Taylor et al., 2022), and one of the largest problems in current LLMs–hallucination–becomes a strength for discovering new molecules (Edwards et al., 2022), proteins (Liu et al., 2023a), and materials (Xie et al., 2023).

Another noteworthy application is NLP for Medicine. As a particular motivating example, there are an estimated $10^{33}$ realistic drug-like molecules (Polishchuk et al., 2013). Within these drugs, there are substructures which confer beneficial drug properties, and the knowledge about these properties are reported in millions of scientific papers. However, existing LLMs are pretrained only from unstructured text and fail to capture this knowledge, in part due to inconsistencies in the literature.

Recent solutions for domain-knowledge-empowered LLMs include development of a lightweight adapter framework to select and integrate structured domain knowledge into LLMs (Lai et al., 2023b), data augmentation for knowledge distillation from LLMs in general domain to scientific domain (Wang et al., 2023g), and tool learning frameworks leveraging foundation models for more complicated sequential actions problem solving (Qin et al., 2023; Qian et al., 2023). Overall, future research can explore bespoke architectures, data acquisition techniques, and training methodologies for comprehending the diverse modalities, domain-specific knowledge, and applications within science.

## 6.3 NLP for Education

LLMs readily capture a vast knowledge of many subjects, and augmenting LLMs with external knowledge naturally leads to improved abilities for eliciting that knowledge to generate lesson plans and materials. However, there are also applications in education which seem distinct from general NLP tasks. In particular, personalizing education and the educational experience with LLMs would allow educators to focus on the more general efforts of high-level teaching. Then, the utility of using language models to educate comes not from the language model's ability to "learn" the appropriate

knowledge but in its ability to find associations. One facet of this challenge comes from identifying and analyzing gaps in a student's understanding or learning. For example, apart from simply scoring essays or responses across discrete dimensions such as fluency or sentence structure or by identifying keyspans (Mathias and Bhattacharyya, 2020; Takano and Ichikawa, 2022; Fiacco et al., 2022), one could use LLMs to determine which parts of a freeform submission indicate a gap and associate it with a learning goal provided by the teacher, without using specific (and costly to create) gold-labeled responses, so that the student has actionable feedback and can work on self-improvement. As part of this work, we need to accurately identify which portions of the response are written by the student as opposed to copied from an AI assistant. This would ensure that gaps aren't hidden, but would require a longitudinal view of the student's ability. Also, we must be able to ensure that the LLM's recommendations are based on actual details of the student and the text rather than being general predictions with high priors or based on hallucinations. Furthermore, rather than simplifying original lesson materials (Mallinson et al., 2022; Omelianchuk et al., 2021), we should invest in using LLMs to generate or retrieve materials or scaffolding that *help* to advance the students' learning rate.

## 7   What We Need

Our overall aim is to combat both the stultification of NLP as a mere evaluation optimization endeavor and to dispel fears that LLMs and generative AI will shut down the field. As an old saying goes, frequent moves make a tree die but a person prosperous. Just as NLP researchers in the 1980s had to learn about machine learning and then embrace it as a core technique in the field, so we now must explore and embrace LLMs and their capabilities. Machine learning did not 'solve' the challenges of NLP: it did not produce an engine that could learn languages, translate, answer questions, create poetry, and do all the things a child can do. Some people claim that LLMs can do all this, and more. But we are in the first flush of engagement, and have not yet have time to discover all their shortcomings.

Central is the challenge of scale. No child needs to read or hear more than half the internet's English text in order to use language. What reasoning and sensory capabilities do people have that LLMs lack? How can NLP research evolve to model and encompass those? We urgently need global infrastructures to dramatically scale up computing resources, because the open-source models still cannot achieve performance comparable to GPT variants (Gudibande et al., 2023). But we also urgently need deeper thinking about the foundational conceptual models driving our field.

During this unique period when NLP researchers feel uncertain regarding which research problems to pursue, we as a community need a collective effort to systematically change and refine our paper review system and academic success measurements, in order to establish a more inclusive research environment and encourage researchers (particularly those in junior positions) to explore long-term, high-risk topics that are crucial for the entire field. The new challenges also require us to be more open-minded to close collaboration with researchers from other fields, including social science, natural science, computer vision, knowledge representation and reasoning, and human-computer interaction.

## Limitations

In this paper we describe some new or under-explored NLP research directions that remain dissertation-worthy. We propose a wider and exciting version of NLP that encourages people to focus on a wider range of more challenging and difficult problems with exciting potential impacts for social good. These problems may not always admit of easy datasets and pure machine learning solutions. Our list is not meant to be exhaustive, and we choose these directions as examples. It is up to NLP researchers to uncover the problems and develop novel solutions.

## Ethical Considerations

The research areas listed in this document are a few of the main areas ripe for exploration; additional ones exist. We do not intend for our proposed positions to be forcefully pedagogical. We encourage diverse and deeper investigation of worthy research areas. Within these proposed directions, we acknowledge that some require access to users' personal information (e.g. chatbot personalization in Section 6.1), and some applications might have high impact on users (e.g. using models to assess a student's grasp of knowledge for targeted education

in Section 6.3). The use of LLMs for creative work has also led to concerns about copyright and regulations over whether AI can be credited as authors. We do not support the use of LLMs for screening or resource allocation purposes without safeguarding measures. Even for lower risk use cases, we opt for more research on the robustness, transparency, and fairness of systems. Finally, we must evaluate the compliance of prompting LLMs with laws and regulations. For instance in education applications, if we require information about the student, we must refer to laws such as FERPA/DPA/GDPR, especially in an online learning setting.

## Acknowledgements

This work is based upon work supported by U.S. DARPA KAIROS Program No. FA8750-19-2-1004, U.S. DARPA CCU Program No. HR001122C0034, U.S. DARPA ECOLE Program No. HR00112390060, U.S. DARPA ITM FA8650-23-C-7316, U.S. DARPA SemaFor Program No. HR001120C0123 and U.S. DARPA INCAS Program No. HR001121C0165. The opinions, views and conclusions contained herein are those of the authors and should not be interpreted as necessarily representing the official policies, either expressed or implied, of DARPA, or the U.S. Government. The U.S. Government is authorized to reproduce and distribute reprints for governmental purposes notwithstanding any copyright annotation therein.

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
