# OpenReview forum: "Defining a New NLP Playground"
_EMNLP/2023/Conference — EMNLP 2023 Findings_

### Official Review · Reviewer_52jY · 2023-07-30

**Typos Grammar Style And Presentation Improvements:** Very well written, no recommendation …
**Soundness:** 5

**Excitement:**

5: Transformative: This paper is likely to change its subfield or computational linguistics broadly. It should be considered for a best paper award. This paper changes the current understanding of some phenomenon, shows a widely held practice to be erroneous in someway, enables a promising direction of research for a (broad or narrow) topic, or creates an exciting new technique.

**Missing References:**

None given the focus of the paper.

**Paper Topic And Main Contributions:**

In view of recent advances in machine learning and LLMs, the authors summarize the current state of NLP and discuss in detail possible research evolutions of NLP when embracing LLMs and their capabilities. They point out a number of deficiencies of LLMs,  and propose topics such as "Theoretical Analysis of LLMs", "Faithfulness and Factuality", "New and Challenging Learning Paradigms",  "Human centered NLP".

**Questions For The Authors:**

A) It is unclear which role you would assign lingustics in the new fields of research that you are outlining. In traditional NLP, linguistic theories have always played the role of providing theoretical background (e.g. about syntactic categories, grammatical structure, etc.). To what extent, do you think, will the new paradigm of LLMs also affect linguistic theory and its impact on NLP?
B) It also is unclear how you would relate foundational models with domain specific LLMs when building question-answering systems for specific applications. Would you deem the traditional distinction between general language and languages for special purposes relevant for your discussion? (If yes, this would lead on to questions about how to identiy domain specific vocabulary and deviations from general language.)

**Reasons To Accept:**

The paper succinctly summarizes deficiencies of LLMs and outlines in detail new research avenues given the recent advancements of LLMs. It thus addresses a central concern of the NLP community of how to position NLP in view of the challenging breakthrough of LLMs.

**Reasons To Reject:**

No reasons to reject the paper.
The paper is well written, well-balanced in considering possible research directions, and fair when characterizing the preconditions of present day LLMs and mainstraim NLP, so that I do not see any reason to reject the paper.

**Reproducibility:**

N/A: Doesn't apply, since the paper does not include empirical results.

**Reviewer Confidence:**

4: Quite sure. I tried to check the important points carefully. It's unlikely, though conceivable, that I missed something that should affect my ratings.

---

> ### Author Rebuttal · Authors · 2023-08-29
>
> We would like to thank you for supporting our paper and for your thoughtful comments.
>
>
> To address your questions:
>
>
>
> A:
> - **LLMs’ impact on linguistic theory**: Many linguistic theories assume a certain level of processing ability for the theory to be fully realized which is exactly what LLMs could provide. We think that LLMs can be seen as powerful statistical engines that can be a tool for verifying and discovering linguistic theories at an unprecedented scale.
>
> - **Linguistic theory’s role on NLP**: While the pretraining of LLMs does not rely on any linguistic guidance, a large part of the extraordinary performance of today’s LLMs is often attributed to their “alignment” to human understanding of language through supervised learning. We believe that linguistic theories could be very valuable in performing systematic diagnosis of LLM behavior and guiding the design of better alignment strategies. In addition, linguistic resources such as PropBank, and FrameNet can still be integrated into LLMs through knowledge augmentation approaches that we discuss in Section 3.1.
>
> B: Although we did not set aside a separate section for the discussion of domain-specific LLMs, this issue does arise in many of the directions that we covered, including Section 3.1 Knowledge Reasoning, 4.2 Online Learning (the motivation for updating LMs is often to incorporate domain-specific data), 6.2 NLP for Science, and 6.3 NLP for Education.
>
> The traditional distinction between general language and language for special purposes (sub-language) assumes that sub-languages are more restricted in the sense of having a restricted domain of reference and limited usage of language's lexical, morphological, syntactic, semantic, and discourse structures [1]. Since modern LLMs are trained on an enormous amount of data, they show good generalization to domains that were previously thought of as sub-languages (weather reports, maintenance manuals).
>
> The challenge for LLMs is primarily in capturing long-tail knowledge: knowledge that does not frequently appear in web text. In addition, for some domain-specific applications, like summarizing medical records, hallucination would be a big issue. A potential solution would be to have a modularized system where a general model routes questions to appropriate domain-specific LLMs. While general models may not be experts, they are quite capable of directing questions to the appropriate expert.
>
> [1] (Richard Kittredge. 1982. Sublanguages. American Journal of Computational Linguistics, 8(2):79–84.)

---

### Official Review · Reviewer_AhjZ · 2023-08-02

**Soundness:** 4

**Excitement:**

3: Ambivalent: It has merits (e.g., it reports state-of-the-art results, the idea is nice), but there are key weaknesses (e.g., it describes incremental work), and it can significantly benefit from another round of revision. However, I won't object to accepting it if my co-reviewers champion it.

**Paper Topic And Main Contributions:**

The contribution of this paper lies in curating a list of research papers that highlight study-worthy research directions that have gained relevance with the advent of Large Language Models.

**Questions For The Authors:**

Could you provide a concise explanation of the main finding you discovered regarding the current direction of NLP research, as revealed through the curation of these research papers?

**Reasons To Accept:**

The paper's comprehensive list of topics can greatly benefit students who are embarking on NLP research, serving as a valuable summary and guiding resource for their studies.

**Reasons To Reject:**

The paper falls short in terms of original contributions, as it primarily offers a mere list of related works and a comprehensive compilation of research areas related to Large Language Models. However, providing only these aspects is insufficient to meet the publication standards for acceptance.

**Reproducibility:**

N/A: Doesn't apply, since the paper does not include empirical results.

**Reviewer Confidence:**

3: Pretty sure, but there's a chance I missed something. Although I have a good feel for this area in general, I did not carefully check the paper's details, e.g., the math, experimental design, or novelty.

---

> ### Author Rebuttal · Authors · 2023-08-29
>
> Thank you for your comments and feedback.  We want to clarify possible misconceptions of our work:
>
> 1) Our work is intended to describe a focused collection of research areas, supported by citations to work in those growing areas. This collection is meant to be inspiring for junior (and senior) researchers, and is not meant to be exhaustive (we select a subset of ideas).  In essence, we believe that this work directly addresses the EMNLP theme track topic of “What are the opportunities LLMs offer to NLP research ?”.
>
> 2) We believe the work goes beyond a simple list, which we agree would be of limited value to the community.  We describe each area thoroughly and the papers that are cited are the most representative ones meant to support the inclusion of the idea in addition to providing pointers for the reader.
>
> Beyond this, we also raise novel questions and provide our own insights (which we welcome the community to stress test thoroughly).  For example, we raised novel questions that are 1)  important for either theoretical understanding of the model or real-world application of the model; 2) new or with newly emerged challenges; 3) feasible for research labs without extensive resources.
>
> In terms of insights, we provide findings about the current gaps in literature, and also point out unexplored questions worthy of future research, which we interlace in each section. To list a few:
> - Line 142-149, 157-162: For emergent abilities, we point out the directions on pursuing new metrics on LLMs’ submissiveness (whether they authentically follow instructions even under adversarial prompts) and the explanation trustworthiness (whether LLMs will justify even wrong answers).
> - Line 178-191: We point out the long-overlooked gap between model-based and model-agnostic theories on LLMs. We also challenge the commonly used mathematical frameworks as not suitable for describing the distribution of language, and advocate for efforts in addressing this concern.
> - Line 238-241: For understanding knowledge with LMs, besides probing for factual knowledge and commonsense, we also need to understand how associations and rules work in LMs (do LMs capture first-order logic in a consistent way or is it just memorizing instances? Can LMs combine multiple rules for inference?)
> - Line 320-330: Current LLMs are not suitable for creative generation out of the box because they seek the maximize the probability of generated sequences, which leads to boring responses. They also have problems maintaining consistency (of characters and background) over long output sequences.
>
> We appreciate the feedback, and If the paper is accepted, we will work to make these contributions clearer to the reader.

---

### Official Review · Reviewer_R4dS · 2023-08-06

**Soundness:** 4

**Excitement:**

4: Strong: This paper deepens the understanding of some phenomenon or lowers the barriers to an existing research direction.

**Paper Topic And Main Contributions:**

The authors present their views on the current state of the NLP domain. New large generative language models have made it possible to solve NLP tasks with much greater efficiency than previous ones. This has changed the domain quite rapidly and has made the future and most interesting directions of NLP research much less predictable. The authors present various areas where they believe more research is needed and is feasible in academia. They characterize the current state of knowledge and the research that has already been done, and point out where further efforts are needed. They point out knowledge acquisition and reasoning, fact-finding, multimodal learning, benchmarks and metrics, among others. They also show the problems associated with NLP on social and scientific grounds.

**Reasons To Accept:**

After the sudden change in the field of NLP, there is a great need to analyze which research directions are the most important/interesting to continue.  The article outlines many different possible areas of exploration

**Reasons To Reject:**

The description of so many issues inevitably had to be short, so there are not many details.

**Reproducibility:**

N/A: Doesn't apply, since the paper does not include empirical results.

**Reviewer Confidence:**

3: Pretty sure, but there's a chance I missed something. Although I have a good feel for this area in general, I did not carefully check the paper's details, e.g., the math, experimental design, or novelty.

---

> ### Author Rebuttal · Authors · 2023-08-29
>
> We would like to thank R1 for appreciating our effort in examining the current state of NLP and promoting promising future directions.
> Given that we would have an extra page of space (and we can make use of the appendix) for publication, we do plan to elaborate more details for each topic and we would be grateful if you could point out some places that you believe needed more elaboration.

---

### Meta-Review · Area_Chair_nsFp · 2023-09-19

**Recommendation:** 4

**Metareview:**

This paper presents a survey on the current state of NLP. The authors characterize the current state of knowledge and the research that has already been done, and point out where further efforts are needed. The paper covers many research areas related to large language models, and as a result, there is a lack of depth. However, it addresses a central concern of the NLP community of how to position NLP in view of the challenging breakthrough of LLMs.

---

### Decision · Program_Chairs · 2023-10-07

**Decision:**

Accept-Findings

**Comment:**

This paper presents a survey on the current state of NLP. The authors characterize the current state of knowledge and the research that has already been done, and point out where further efforts are needed. The paper covers many research areas related to large language models, and as a result, there is a lack of depth. However, it addresses a central concern of the NLP community of how to position NLP in view of the challenging breakthrough of LLMs.